# Improving the Improved Training of Wasserstein GANs: A Consistency Term and Its Dual Effect

**Xiang Wei**[1,2*]**, Boqing Gong**[3*]**, Zixia Liu**[1]**, Wei Lu**[2]**, Liqiang Wang**[1]
[1]Department of Computer Science, University of Central Florida, Orlando, FL, USA 32816
[2]School of Software Engineering, Beijing Jiaotong University, Beijing, China 100044
[3]Tencent AI Lab, Bellevue, WA, USA 98004
`yqweixiang@knights.ucf.edu, boqinggo@outlook.com`
`zixia@knights.ucf.edu, luwei@bjtu.edu.cn, lwang@cs.ucf.edu`

## Abstract

Despite being impactful on a variety of problems and applications, the generative adversarial nets (GANs) are remarkably difficult to train. This issue is formally analyzed by Arjovsky & Bottou (2017), who also propose an alternative direction to avoid the caveats in the minmax two-player training of GANs. The corresponding algorithm, called Wasserstein GAN (WGAN), hinges on the 1-Lipschitz continuity of the discriminator. In this paper, we propose a novel approach to enforcing the Lipschitz continuity in the training procedure of WGANs. Our approach seamlessly connects WGAN with one of the recent semi-supervised learning methods. As a result, it gives rise to not only better photo-realistic samples than the previous methods but also state-of-the-art semi-supervised learning results. In particular, our approach gives rise to the inception score of more than 5.0 with only 1,000 CIFAR-10 images and is the first that exceeds the accuracy of 90% on the CIFAR-10 dataset using only 4,000 labeled images, to the best of our knowledge.

## 1 Introduction

We have witnessed a great surge of interests in deep generative networks in recent years (Kingma & Welling, 2013; Goodfellow et al., 2014; Li et al., 2015). The central idea therein is to feed a random vector to a (*e.g.*, feedforward) neural network and then take the output as the desired sample. This sampling procedure is very efficient without the need of any Markov chains.

In order to train such a deep generative network, two broad categories of methods are proposed. The first is to use stochastic variational inference (Kingma & Welling, 2013; Rezende et al., 2014; Kingma et al., 2014) to optimize the lower bound of the data likelihood. The other is to use the samples as a proxy to minimize the distribution divergence between the model and the real through a two-player game (Goodfellow et al., 2014; Salimans et al., 2016), maximum mean discrepancy (Li et al., 2015; Dziugaite et al., 2015; Li et al., 2017b), f-divergence (Nowozin et al., 2016; Nock et al., 2017), and the most recent Wasserstein distance (Arjovsky et al., 2017; Gulrajani et al., 2017).

With no doubt, the generative adversarial networks (GANs) among them (Goodfellow et al., 2014) have the biggest impact thus far on a variety of problems and applications (Radford et al., 2015; Denton et al., 2015; Im et al., 2016; Isola et al., 2016; Springenberg, 2015; Sutskever et al., 2015; Odena, 2016; Zhu et al., 2017). GANs learn the generative network (generator) by playing a two-player game between the generator and an auxiliary discriminator network. While the generator has no difference from other deep generative models in the sense that it translates a random vector into a desired sample, it is impossible to calculate the sample likelihood from it. Instead, the discriminator serves to evaluate the quality of the generated samples by checking how difficult it is to differentiate them from real data points.

---

*Equal contribution.

However, it is remarkably difficult to train GANs without good heuristics (Goodfellow et al., 2014; Salimans et al., 2016; Radford et al., 2015) which may not generalize across different network architectures or application domains. The training dynamics are often unstable and the generated samples could collapse to limited modes. These issues are formally analyzed by Arjovsky & Bottou (2017), who also propose an alternative direction (Arjovsky et al., 2017) to avoid the caveats in the minmax two-player training of GANs. The corresponding algorithm, namely, Wasserstein GAN (WGAN), shows not only superior performance over GANs but also a nice correlation between the sample quality and the value function that GANs lack.

## 1.1 BACKGROUND: WGAN AND THE IMPROVED TRAINING OF WGAN

WGAN (Arjovsky et al., 2017) aims to learn the generator network $G(\boldsymbol{z})$, for any random vector $\boldsymbol{z} \sim \mathbb{P}_z$, such that the Wasserstein distance is minimized between the resulting distribution $\mathbb{P}_G$ of the generated samples $\{G(\boldsymbol{z})\}$ and the real distribution $\mathbb{P}_r$ underlying the observed data points $\{\boldsymbol{x}\}$; in other words, $\min_G W(\mathbb{P}_r, \mathbb{P}_G)$. The Wasserstein distance $W(\mathbb{P}_r, \mathbb{P}_G)$ is shown a more sensible cost function for learning the distributions supported by low-dimensional manifolds than the other popular distribution divergences and distances — for example, the Jensen-Shannon (JS) divergence implicitly employed in GANs (Goodfellow et al., 2014).

Due to the Kantorovich-Rubinstein duality (Villani, 2008) for calculating the Wasserstein distance, the value function of WGAN is then written as

$$\min_G \max_{D \in \mathcal{D}} \mathbb{E}_{\boldsymbol{x} \sim \mathbb{P}_r}\big[D(\boldsymbol{x})\big] - \mathbb{E}_{\boldsymbol{z} \sim \mathbb{P}_z}\big[D(G(\boldsymbol{z}))\big], \tag{1}$$

where $\mathcal{D}$ is the set of 1-Lipschitz functions. Analogous to GANs, we still call $D$ the "discriminator" although it is actually a real-valued function and not a classifier at all. Arjovsky et al. (2017) specify this family of functions $\mathcal{D}$ by neural networks and then use weight clipping to enforce the Lipschitz continuity. However, as the authors note, the networks' capacities become limited due to the weight clipping and there could be gradient vanishing problems in the training.

**Improved training of WGAN.** Gulrajani et al. (2017) give more concrete examples to illustrate the perils of the weight clipping and propose an alternative way of imposing the Lipschitz continuity. In particular, they introduce a gradient penalty term by noting that the differentiable discriminator $D(\cdot)$ is 1-Lipschitz if and only if the norm of its gradients is at most 1 *everywhere*,

$$GP|_{\widehat{\boldsymbol{x}}} := \mathbb{E}_{\widehat{\boldsymbol{x}}}\Big[\big(\|\nabla_{\widehat{\boldsymbol{x}}} D(\widehat{\boldsymbol{x}})\|_2 - 1\big)^2\Big] \tag{2}$$

where $\widehat{\boldsymbol{x}}$ is uniformly sampled from the straight line between a pair of data points sampled from the model $\mathbb{P}_G$ and the real $\mathbb{P}_r$, respectively. A similar regularization is used by Kodali et al. (2017).

**Potential caveats.** Unlike the weight clipping, however, by no means one can penalize *everywhere* using this term through a finite number of training iterations. As a result, the gradient penalty term $GP$ takes effect only upon the sampled points $\widehat{\boldsymbol{x}}$, leaving significant parts of the support domain not examined at all. In particular, consider the observed data points and their underlying manifold that supports the real distribution $\mathbb{P}_r$. At the beginning of the training stage, the generated sample $G(\boldsymbol{z})$ and hence $\widehat{\boldsymbol{x}}$ could be distant from the manifold. The Lipschitz continuity over the manifold is not enforced until the generative model $\mathbb{P}_G$ becomes close enough to the real one $\mathbb{P}_r$, if it can.

## 1.2 OUR APPROACH AND CONTRIBUTIONS

In light of the above pros and cons, we propose to improve the improved training of WGAN by additionally laying the Lipschitz continuity condition over the manifold of the real data $\boldsymbol{x} \sim \mathbb{P}_r$. Moreover, instead of focusing on one particular data point at a time, we devise a regularization over a pair of data points drawn near the manifold following the most basic definition of the 1-Lipschitz continuity. In particular, we perturb each real data point $\boldsymbol{x}$ twice and use a Lipschitz constant to bound the difference between the discriminator's responses to the perturbed data points $\boldsymbol{x}', \boldsymbol{x}''$.

Figure 1 illustrates our main idea. The gradient penalty $GP|_{\widehat{\boldsymbol{x}}}$ often fails to check the continuity of region near the real data $\boldsymbol{x}$, around which the discriminator function can freely violate the 1-Lipschitz continuity. We alleviate this issue by explicitly checking the continuity condition using the two perturbed version $\boldsymbol{x}', \boldsymbol{x}''$ near any observed real data point $\boldsymbol{x}$.

In this paper, we make the following contributions. (1) We propose an alternative mechanism for enforcing the Lipschitz continuity over the family of discriminators by resorting to the basic definition of the Lipschitz continuity. It effectively improves the gradient penalty method (Gulrajani et al., 2017) and gives rise to generators with more photo-realistic samples and higher inception scores (Salimans et al., 2016). (2) Our approach is very data efficient in terms of being less prone to overfitting even for very small training sets. We do not observe obvious overfitting phenomena even when the model is trained on only 1000 images of CIFAR-10 (Krizhevsky & Hinton, 2009). (3) Our approach can be seamlessly integrated with GANs to be a competitive semi-supervised training technique (Chapelle et al., 2009) thanks to that both inject noise to the real data points.

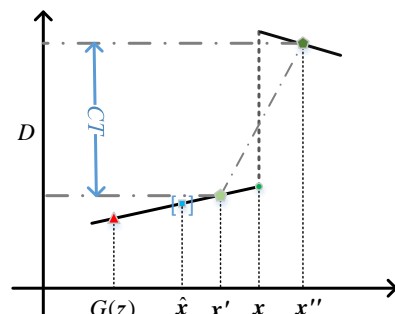

Figure 1: Illustration of our main idea. In addition to the gradient penalty over $\widehat{x}$, we also examine $x'$ and $x''$ around the real data point $x$ in each iteration.

As results, we are able to report the state-of-the-art results on the generative model with a inception score of $8.81 \pm 0.13$ on CIFAR-10, and the semi-supervised learning results of $9.98 \pm 0.21$ on CIFAR-10 using only 4,000 labeled images — especially, they are significantly better than all the existing GAN-based semi-supervised learning results, to the best of our knowledge.

## 2 APPROACH

We firstly review the definition of the Lipschitz continuity and then discuss how to use it to regularize the training of WGAN. We then arrive at an approach that can be seamlessly integrated with the semi-supervised learning method (Laine & Aila, 2016). By bringing the best of the two worlds, we report better semi-supervised learning results than both (Laine & Aila, 2016) and existing GAN-based methods.

### 2.1 IMPROVING THE IMPROVED TRAINING OF WGAN

Let $d$ denote the $\ell_2$ metric on an input space used in this paper. A discriminator $D : \mathcal{X} \mapsto \mathcal{Y}$ is Lipschitz continuous if there exists a real constant $M \geqslant 0$ such that, for all $x_1, x_2 \in \mathcal{X}$,

$$d(D(x_1), D(x_2)) \leqslant M \cdot d(x_1, x_2). \tag{3}$$

Immediately, we can add the following soft consistency term ($CT$) to the value function of WGAN in order to penalize the violations to the inequality in eq. (3),

$$CT|_{x_1, x_2} = \mathbb{E}_{x_1, x_2} \left[ \max \left( 0, \frac{d(D(x_1), D(x_2))}{d(x_1, x_2)} - M' \right) \right] \tag{4}$$

**Remarks.** Here we face the same snag as in (Gulrajani et al., 2017), *i.e.*, it is impractical to substitute all the possibilities of $(x_1, x_2)$ pairs into eq. (4). What pairs and which regions of the input set $\mathcal{X}$ should we check for eq. (4)? Arguably, it is fairly safe to limit our scope to the manifold that supports the real data distribution $\mathbb{P}_r$ and its surrounding regions mainly for two reasons. First, we keep the gradient penalty term and improve it by the proposed consistency term in our overall approach. While the former enforces the continuity over the points sampled between the real and generated points, the latter complement the former by focusing on the region around the real data manifold instead. Second, the distribution of the generative model $\mathbb{P}_G$ is virtually desired to be as close as possible to $\mathbb{P}_r$. We use the notation $M$ in eq. (3) and a different $M'$ in eq. (4) to reflect the fact that the continuity will be checked only sparsely at finite data points in practice.

**Perturbing the real data.** To this end, the very first version we tried was to directly add Gaussian noise $\delta$ to each real data point, resulting in a pair of $x + \delta_1, x + \delta_2$, where $x \sim \mathbb{P}_r$. However, as noted by Arjovsky et al. (2017) and Wu et al. (2016), we found that the samples from the generator become blurry due to the Gaussian noise used in the training. We have also tested the dropout noise that is applied to the input and found that the resulting MNIST samples are cut off here and there.

---

**Algorithm 1** our proposed **CT-GAN** for training a generative neural net. Most of our experiments are conducted with the default values $\lambda_1 = 10, \lambda_2 = 2$, $N = 5$, $\gamma = 0.0002$, $b = 64$, $iter = 10^6$

---

**Require:** $b$, the batch size. $\lambda_1, \lambda_2$, weights. $\gamma$, the Learning rate. $iter$, number of iterations.
 1: **for** $iter$ of training iterations **do**
 2:     **for** $N$ of iterations **do**
 3:         **for** $i = 1, ..., b$ **do**
 4:             Sample real data $\boldsymbol{x} \sim \mathbb{P}_r$, random vector $\boldsymbol{z} \sim \mathbb{P}_z$, and a random number $\epsilon \sim U[0, 1]$.
 5:             $\hat{x} \leftarrow \epsilon \boldsymbol{x} + (1 - \epsilon) G(\boldsymbol{z})$
 6:             $L^{(i)} \leftarrow D(G(\boldsymbol{z})) - D(\boldsymbol{x}) + \lambda_1 GP|_{\widehat{\boldsymbol{x}}} + \lambda_2 CT|_{\boldsymbol{x}', \boldsymbol{x}''}$
 7:         **end for**
 8:         $\theta_d \leftarrow Adam(\nabla_{\theta_d} \frac{1}{b} \sum_{i=1}^{b} L^{(i)}, \theta_d, \gamma)$
 9:     **end for**
10:     Sample $b$ random vectors $\left\{ \boldsymbol{z}^{(i)} \right\}$
11:     $\theta_g \leftarrow Adam(\nabla_{\theta_g} \frac{1}{b} \sum_{i=1}^{b} -D(G(\boldsymbol{z}^{(i)})), \theta_g, \gamma)$
12: **end for**

---

The success comes after we perturb the hidden layers of the discriminator using dropout, as opposed to the input $\boldsymbol{x}$. When the dropout rate is small, the perturbed discriminator's output can be considered as the output of the clean discriminator in response to a "virtual" data point $\boldsymbol{x}'$ that is not far from $\boldsymbol{x}$. Thus, we denote by $D(\boldsymbol{x}')$ the discriminator output after applying dropout to its hidden layers. In the same manner, we find the second virtual point $\boldsymbol{x}''$ around $\boldsymbol{x}$ by applying the (stochastic) dropout again to the hidden layers of the discriminator, and denote by $D(\boldsymbol{x}'')$ the corresponding output.

Note that, however, it becomes impossible to compute the distance $d(\boldsymbol{x}', \boldsymbol{x}'')$ between the two virtual data points. In this work, we assume it is bounded by a constant and absorb the constant to $M'$. Accordingly, we tune $M'$ in our experiments to take account of this unknown constant; the best results are obtained between $M' = 0$ and $M' = 0.2$. For consistency, we use $M' = 0$ to report all the results in this paper.

**A consistency regularization.** Our final consistency regularization takes the following form,

$$CT|_{\boldsymbol{x}', \boldsymbol{x}''} = \mathbb{E}_{\boldsymbol{x} \sim \mathbb{P}_r} \left[ \max\left( 0, d\left( D(\boldsymbol{x}'), D(\boldsymbol{x}'') \right) + 0.1 \cdot d\left( D_-(\boldsymbol{x}'), D_-(\boldsymbol{x}'') \right) - M' \right) \right] \quad (5)$$

where $D(\boldsymbol{x}')$ is the output of the discriminator given the input $\boldsymbol{x}$ and after we apply dropout to the hidden layers of the discriminator. We envision this is equivalent to passing a "virtual" data point $\boldsymbol{x}'$ through the clean discriminator. We find that it slightly improves the performance by further controlling the second-to-last layer $D_-(\cdot)$ of the discriminator, *i.e.*, the $d\left( D_-(\boldsymbol{x}'), D_-(\boldsymbol{x}'') \right)$ above.

This new consistent regularization $CT|_{\boldsymbol{x}', \boldsymbol{x}''}$ enforces the Lipschitz continuity over the data manifold and its surrounding regions, effectively complementing and improving the gradient penalty $GP|_{\widehat{\boldsymbol{x}}}$ used in the improved training of WGAN. Putting them together, our new objective function for updating the weigts of the discriminator is

$$L = \mathbb{E}_{\boldsymbol{z} \sim \mathbb{P}_z} \left[ D(G(\boldsymbol{z})) \right] - \mathbb{E}_{\boldsymbol{x} \sim \mathbb{P}_r} \left[ D(\boldsymbol{x}) \right] + \lambda_1 GP|_{\widehat{\boldsymbol{x}}} + \lambda_2 CT|_{\boldsymbol{x}', \boldsymbol{x}''}. \quad (6)$$

**Algorithm 1** shows the complete algorithm for learning a WGAN in this paper. For the hyper-parameters, we borrow $\lambda_1 = 10$ from Gulrajani et al. (2017) and use $\lambda_2 = 2$ for all our experiments no matter on which dataset. Another hyper-parameter is $M'$ in eq. (5). As stated previously, $M'$ taking values between 0 and 0.2 gives rise to about the same results in our experiments.

## 2.2 A SEAMLESS CONNECTION WITH A SEMI-SUPERVISED LEARNING METHOD

In this section, we extend the WGAN to a semi-supervised learning approach by drawing insights from two related works.

- Following (Salimans et al., 2016), we modify the output layer of the discriminator such that it has $K + 1$ output neurons, where $K$ is the number of classes of interest and the $(K + 1)$-th neuron is reserved for contrasting the generated samples with the real data using the Wasserstein distance in the WGAN. We use a $(K + 1)$-way softmax as the activation function of the last layer.

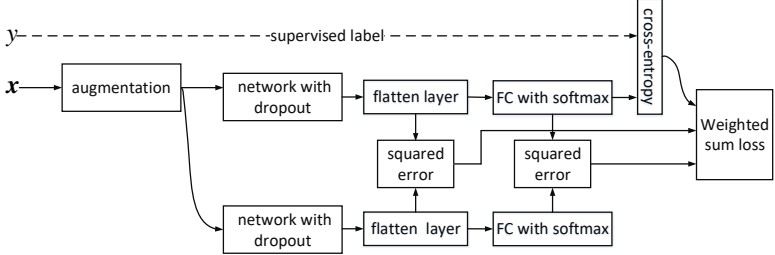

Figure 2: Framework for the semi-supervised training. For clarity, we have omitted the generator.

- Following (Laine & Aila, 2016), we add our consistency regularization $CT|_{\boldsymbol{x}',\boldsymbol{x}''}$ to the objective function of the semi-supervised training, in order to take advantage of the effect of temporal ensembling.

Figure 2 shows the framework for training the discriminator, with the objective function below,

$$
\begin{aligned}
L_{\text{semi\_dis}} = & - \mathbb{E}_{\boldsymbol{x},y \sim \mathbb{P}_{\text{x,y}}} \left[ \log D(y|\boldsymbol{x}) \right] - \mathbb{E}_{\boldsymbol{z} \sim \mathbb{P}_z} \left[ \log D(K+1|G(\boldsymbol{z})) \right] \\
& - \mathbb{E}_{\boldsymbol{x} \sim \mathbb{P}_r} \left[ \log(1 - D(K+1|\boldsymbol{x})) \right] + \lambda CT|_{\boldsymbol{x}',\boldsymbol{x}''},
\end{aligned}
\tag{7}
$$

where the first three terms are the same as in (Salimans et al., 2016), while the last consistency regularization is calculated after we apply dropout to the discriminator. The last term essentially leads to a temporal self-ensembling scheme to benefit the semi-supervised learning. Please see (Laine & Aila, 2016) for more insightful discussions about it.

The generator loss matches the expected features of the generated sample and the real data points,

$$
L_{\text{semi\_gen}} = \left\| \mathbb{E}_{\boldsymbol{z} \sim \mathbb{P}_z}(D_-(G(\boldsymbol{z})) - \mathbb{E}_{\boldsymbol{x} \sim \mathbb{P}_r}(D_-(\boldsymbol{x})) \right\|_2^2.
\tag{8}
$$

## 3 EXPERIMENTAL RESULTS

We conduct experiments on the prevalent MNIST (LeCun et al., 1998), and CIFAR-10 (Krizhevsky & Hinton, 2009) datasets. The code is available on `https://github.com/biuyq/CT-GAN` to facilitate the reproducibility of our results.

### 3.1 MNIST

The MNIST dataset provides 70,000 handwritten digits in total and 10,000 of them are often left out for the testing purpose. Following (Gulrajani et al., 2017), we use only 1,000 of them to train the WGAN for a fair comparison with it. We use all the 60,000 training examples in the semi-supervised learning experiments and reveal the labels of 100 of them (10 per class). This setup is the same as in (Rasmus et al., 2015). No data augmentation is used. Please see Appendix A for the network architectures of the generator and discriminator, respectively.

**Qualitative results.** Figure 3 shows the generated samples with improved training of WGAN by the gradient penalty (GP-WGAN) and ours with the consistency regularization (CT-GAN), respectively, after 50,000 generator iterations. It is clear that our approach gives rise to more realistic samples than GP-WGAN. The contrasts of our samples between the foreground and the background are in general sharper than those of GP-WGAN.

**Overfitting.** We find that our approach is less prone to overfitting. To demonstrate this point, we show the convergence curves of the discriminator's value functions by GP-WGAN and our CT-GAN in Figure 4. The red curves are evaluated on the training set and the blue ones are on the test set. We can see that the results on the test set become saturated pretty early in GP-WGAN, while ours can consistently decrease the costs on both the training and the test sets. This observation also holds for the CIFAR-10 dataset (cf. Appendix E).

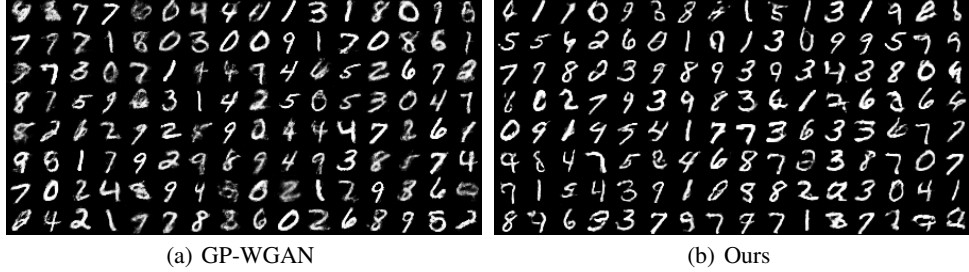

(a) GP-WGAN           (b) Ours

Figure 3: Images generated by (a) GP-WGAN (Gulrajani et al., 2017) and (b) Our CT-GAN, respectively.

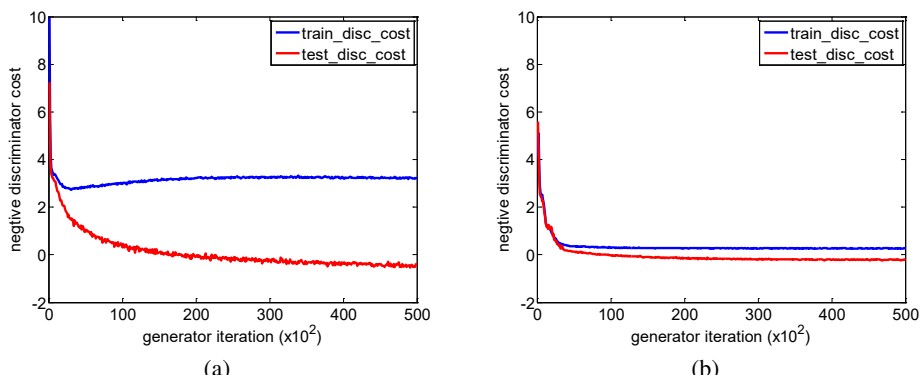

(a)                (b)

Figure 4: Convergence curves of the discriminator cost: (a) GP-WGAN and (b) Our CT-GAN.

**Semi-supervised learning results.** We compare our semi-supervised learning results with those of several competitive methods in Table 1. Our approach is among the best on this MNIST dataset.

## 3.2 CIFAR-10

CIFAR-10 (Krizhevsky & Hinton, 2009) contains 50,000 natural images of size $32 \times 32$. We use it to test two networks for the generative model: a small CNN and a ResNet (cf. Appendix A for the network structures). For the former we use only 1,000 images to train the model and we use the whole training set to learn the ResNet.

**Qualitative results.** Figure 5 contrasts the samples we generated to those by GP-WGAN when the generator is the small CNN. Figure 7 shows the results by a larger-scale ResNet. Our results are more photo-realistic.

Additionally, we also draw the histograms of the discriminator's weights in Figure 6 after we train it using GP-WGAN and CT-GAN, respectively. It is interesting to see that ours controls the weights within a smaller and more symmetric range $[-0.67, 0.96]$ than the $[-2.00, 10.12]$ by GP-WGAN, partially explaining why our approach is less prone to overfitting.

Table 1: Comparing our semi-supervised learning approach with state-of-the-art ones on MNIST.

| Method | Test error (%) |
|---|---|
| Ladder (Rasmus et al., 2015) | 1.06±0.37 |
| VAT (Miyato et al., 2017) | 1.36 |
| CatGAN (Springenberg, 2015) | 1.39±0.28 |
| Improved GAN (Salimans et al., 2016) | $0.93 \pm 0.065$ |
| Triple GAN (Li et al., 2017a) | $0.91 \pm 0.58$ |
| **Our CT-GAN** | **$0.89 \pm 0.13$** |

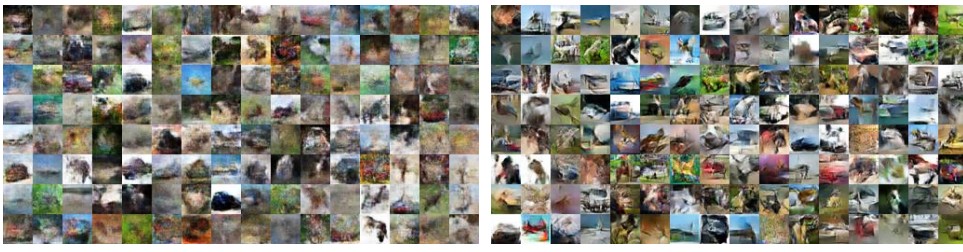

(a) GP-WGAN (inception score: $2.98 \pm 0.11$)      (b) Our CT-GAN (inception score: $5.13 \pm 0.12$)

Figure 5: Generated samples by (a) GP-WGAN and (b) our CT-GAN. Here the generator is a small CNN. See Figure 7 for the samples by a ResNet.

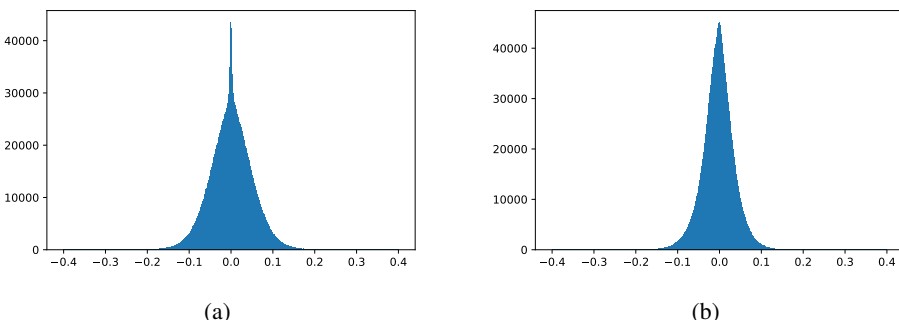

(a)            (b)

Figure 6: Histograms of the weights of the discriminators trained by (a) GP-WGAN and (b) CT-GAN, respectively.

Table 2: Inception score and accuracy of different models on CIFAR-10

| Method | Supervised IS | Unsupervised IS | Accuracy(%) |
|---|---|---|---|
| SteinGANs (Wang & Liu, 2016) | 6.35 | – | – |
| DCGANs (Radford et al., 2015) | 6.58 | $6.16 \pm 0.07$ | – |
| Improved GANs (Salimans et al., 2016) | $8.09 \pm 0.07$ | – | – |
| AC-GANs (Odena et al., 2016) | $8.25 \pm 0.07$ | – | – |
| GP-WGAN (Gulrajani et al., 2017) | $8.42 \pm 0.10$ | $7.86 \pm 0.07$ | 91.85 |
| SGANs (Huang et al., 2016) | $8.59 \pm 0.12$ | – | – |
| ALI (Warde-Farley & Bengio, 2016) | – | $5.34 \pm 0.05$ | – |
| BEGAN (Berthelot et al., 2017) | – | 5.62 | – |
| EGAN-Ent-VI (Dai et al., 2017) | – | $7.07 \pm 0.10$ | – |
| DFM (Warde-Farley & Bengio, 2016) | – | $7.72 \pm 0.07$ | – |
| Our CT-GAN | **8.81±0.13** | **8.12±0.12** | **95.91** |

**Comparison of the inception scores.** Finally, we compare our approach with GP-WGAN on the whole training set for both unsupervised and supervised generative-purpose task using ResNet. For model selection, we use the first 50,000 samples to compute the inception scores (Salimans et al., 2016), then choose the best model, and finally report the "test" score on another 50,000 samples. The experiment follows the previous setup in (Odena et al., 2016). From the comparison results in Tables 2, we conclude that our proposed model achieves the highest inception score on the CIFAR-10 dataset, to the best of our knowledge. Some generated samples are shown in Figure 7.

For the small CNN based generator, the inception scores of GP-WGAN and our CT-GAN are $2.98 \pm 0.11$ and $5.13 \pm 0.12$, respectively.

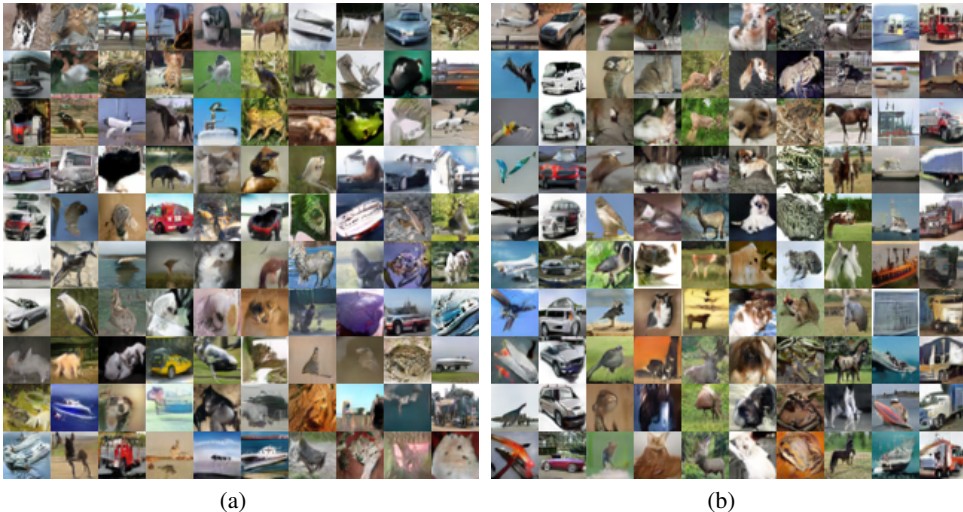

(a)                                                      (b)

Figure 7: Generated samples by our ResNet model: (a) Generated samples by unsupervised model and (b) Generated samples by supervised model. Each column corresponds to one class in the CIFAR-10 dataset.

Table 3: Comparing our semi-supervised learning approach with state-of-the-art ones on CIFAR-10

| Method | Test error (%) |
|---|---|
| Ladder (Rasmus et al., 2015) | $20.40 \pm 0.47$ |
| VAT (Miyato et al., 2017) | 10.55 |
| TE (Laine & Aila, 2016) | $12.16 \pm 0.24$ |
| Teacher-Student (Tarvainen & Valpola, 2017) | $12.31 \pm 0.28$ |
| CatGANs (Springenberg, 2015) | $19.58 \pm 0.58$ |
| Improved GANs (Salimans et al., 2016) | $18.63 \pm 2.32$ |
| ALI (Dumoulin et al., 2016) | $17.99 \pm 1.62$ |
| CLS-GAN (Qi, 2017) | $17.30 \pm 0.50$ |
| Triple GAN (Li et al., 2017a) | $16.99 \pm 0.36$ |
| Improved semi-GAN (Kumar et al., 2017) | $16.78 \pm 1.80$ |
| Our CT-GAN | $\mathbf{9.98 \pm 0.21}$ |

**Semi-supervised learning.** For the semi-supervised learning approach, we follow the standard training/test split of the dataset but use only 4,000 labels in the training. A regular data augmentation with flipping the images horizontally and randomly translating the images within [-2,2] pixels is used in our paper (No ZCA whitening). We report the semi-supervised learning results in Table 3. The mean and standard errors are obtained by running the experiments 5 rounds. Comparing to several very competitive methods, ours is able to achieve state-of-the-art results. Notably, our CT-GAN outperfroms all the GAN based methods by a large margin. Please see Appendix A for the network architectures and Appendix C for the ablation study of our algorithm.

## 4 CONCLUSION

In this paper, we present a consistency term derived from Lipschitz inequality, which boosts the performance of GANs model. The proposed term has been demonstrated to be an efficient manner to ease the over-fitting problem when data amount is limited. Experiments show that our model obtains the state-of-the-art accuracy and Inception score on CIFAR-10 dataset for both the semi-supervised learning task and the learning of generative models.

**Acknowledgements.** This work is partially supported by NSF IIS-1741431, IIS-1566511, and ONR N00014-18-1-2121. B.G. and L.W. would also like to thank Adobe Research and NVIDIA, and Amazon, respectively, for their gift donations.

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

# Appendices

## APPENDIX A    NETWORK ARCHITECTURES

Table 4 (MNIST) and Table 5 (CIFAR-10) detail the network architectures used in our classification-purpose CT-GAN, where the classifiers are the same as the widely used ones in most semi-supervised networks (Laine & Aila, 2016) except that we apply weight-norm rather than batch-norm. For the generators, we follow the network structures in GP-WGAN (Salimans et al., 2016), but we use lower dimensional noise (50D) as the input to the generator for CIFAR-10 in order not to reproduce the complicated images and instead shift the focus of the training to the classifier.

Table 4: Networks for semi-supervised learning on MNIST

| Classifier C | Generator G |
|---|---|
| Input: Labels $y$, 28*28 Images $x$, | Input: Noise 100 $z$ |
| Gaussian noise 0.3, MLP 1000, ReLU | MLP 500, Softplus, Batch norm |
| Gaussian noise 0.5, MLP 500, ReLU | MLP 500, Softplus, Batch norm |
| Gaussian noise 0.5, MLP 250, ReLU | MLP 784, Sigmoid, Weight norm |
| Gaussian noise 0.5, MLP 250, ReLU | |
| Gaussian noise 0.5, MLP 250, ReLU | |
| Gaussian noise 0.5, MLP 10, Softmax | |

Table 5: Networks for semi-supervised learning on CIFAR-10

| Classifier C | Generator G |
|---|---|
| Input: Labels $y$, 32*32*3 Colored Image $x$, | Input: Noise 50 $z$ |
| 0.2 Dropout | MLP 8192, ReLU, Batch norm |
| 3*3 conv. 128, Pad =1, Stride =1, lReLU, Weight norm | Reshape 512*4*4 |
| 3*3 conv. 128, Pad =1, Stride =1, lReLU, Weight norm | 5*5 deconv. 256*8*8, |
| 3*3 conv. 128, Pad =1, Stride =2, lReLU, Weight norm | ReLU, Batch norm |
| 0.5 Dropout | |
| 3*3 conv. 256, Pad =1, Stride =1, lReLU, Weight norm | |
| 3*3 conv. 256, Pad =1, Stride =1, lReLU, Weight norm | 5*5 deconv. 128*16*16, |
| 3*3 conv. 256, Pad =1, Stride =2, lReLU, Weight norm | ReLU, Batch norm |
| 0.5 Dropout | |
| 3*3 conv. 512, Pad =0, Stride =1, lReLU, Weight norm | |
| 3*3 conv. 256, Pad =0, Stride =1, lReLU, Weight norm | 5*5 deconv. 3*32*32, |
| 3*3 conv. 128, Pad =0, Stride =1, lReLU, Weight norm | Tanh, Weight norm |
| Global pool | |
| MLP 10, Weight norm, Softmax | |

## APPENDIX B    HYPER-PARAMETERS AND OTHER TRAINING DETAILS

For the semi-supervised learning experiments, we set $\lambda = 1.0$ in Eq.(7) in all our experiments. For CIFAR-10, the number of training epochs is set to 1,000 with a constant learning rate of 0.0003. For

Table 6: Generative model for MNIST

| Discriminator | Generator |
|---|---|
| Input: 1*28*28 Image $x$ | Input: Noise $z$ 128 |
| 5*5 conv. 64, Pad = same, Stride = 2, lReLU | MLP 4096, ReLU |
| 0.5 Dropout | Reshape 256*4*4 |
| 5*5 conv. 128, Pad = same, Stride = 2, lReLU | 5*5 deconv. 128*8*8 |
| 0.5 Dropout | ReLU, Cut 128*7*7 |
| 5*5 conv. 256, Pad = same, Stride = 2, lReLU | 5*5 deconv. 64*14*14 |
| 0.5 Dropout | ReLU |
| Reshape 256*4*4 (D_) | 5*5 deconv. 1*28*28 |
| MLP 1 (D) | Sigmoid |

Table 7: Generative model for CIFAR-10

| Discriminator | Generator |
|---|---|
| Input: 3*32*32 Image $x$, | Input: Noise $z$ 128 |
| 5*5 conv. 128, Pad = same, Stride = 2, lReLU | MLP 8192, ReLU, Batch norm |
| 0.5 Dropout | Reshape 512*4*4 |
| 5*5 conv. 256, Pad = same, Stride = 2, lReLU | 5*5 deconv. 256*8*8 |
| 0.5 Dropout | ReLU, Bach norm |
| 5*5 conv. 512, Pad = same Stride = 2, lReLU | 5*5 deconv. 128*16*16 |
| 0.5 Dropout | ReLU, Batch norm |
| Reshape 512*4*4 (D_) | 5*5 deconv. 3*32*32 |
| MLP 1 (D) | Tanh |

MNIST, the number of training epochs is set to 300 with a constant learning rate of 0.003. The other hyper-parameters are exactly the same as in the improved GAN (Salimans et al., 2016).

For the experiments on the generative models, to have a fair comparison for our results with the existing ones, we keep the network structure and hyper-parameters the same as in the improved training of WGAN (Gulrajani et al., 2017) except that we add three *dropout* layers to some of the hidden layers as shown in tables 6 to 8.

## APPENDIX C   ABLATION STUDY OF OUR APPROACH TO SSL

We run an ablation study of our approach to the semi-supervised learning (SSL). Thanks to the dual effect of the proposed consistency (CT) term, we are able to connect GAN with the temporal ensembling (TE) Laine & Aila (2016) method for SSL. Our superior results thus benefit from both of them, verified by the ablation study detailed in Table 9.

Table 8: ResNet for CIFAR-10

| Discriminator | Generator |
|---|---|
| Input: 3*32*32 Image $x$ | Input: Noise $bmz$ 128 |
| [3*3]*2 Residual Block, Resample = DOWN | MLP 2048 |
| 128*16*16 | Reshape 128*4*4 |
| [3*3]*2 Residual Block, Resample = DOWN | [3*3]*2 Residual Block, Resample = UP |
| 128*8*8 0.2 Dropout | 128*8*8 |
| [3*3]*2 Residual Block, Resample = None | [3*3]*2 Residual Block, Resample = UP |
| 128*8*8 0.5 Dropout | 128*16*16 |
| [3*3]*2 Residual Block, Resample = None | [3*3]*2 Residual Block, Resample = UP |
| 128*8*8 0.5 Dropout | 128*32*32 |
| ReLU, Global mean pool (D_) | 3*3 conv. 3*32*32 |
| MLP 1 (D) | Tanh |

If we remove the CT term, the test error goes up to $14.98$, signifying the effectiveness of the CT regularization.

If we remove GAN from our approach, it almost reduces to TE; in fact, all the settings here are the same as TE except that we use an extra regularization ($D_-(.,.)$ in CT) over the second-to-last layer. We can see that the error is still significantly larger than our overall method.

We use the weight normalization as in Salimans et al. (2016), which becomes a core constituent of our approach. The batch normalization would actually invalidate the feature matching in Salimans et al. (2016).

Finally, we also test the version without the regularization to the second-to-last layer and observe a little drop in the performance.

Table 9: Ablation study of our semi-supervised learning method

| Method | Test Error |
|---|---|
| w/o CT | $14.98\pm0.43$ |
| w/o GAN | $11.98\pm0.32$ |
| w batch norm | – |
| w/o $D_-(.,.)$ over the second-to-last layer | $10.70\pm0.24$ |
| Ours | $9.98\pm0.21$ |

## APPENDIX D  EXAMINING THE 1-LIPSCHITZ CONTINUITY

**Norm of gradient.**  In our experiments, we find that although the GP-WGAN (Gulrajani et al., 2017) has applied a Lipschitz constraint in the form of the gradient penalty over the input sampled between a real data point and a generated one, the actual effect on the $\ell_2$ norm of the gradient is not as good as our CT-GAN model in the real data points, as illustrated in Figure 1. We empirically verify this fact by Figure 8, which shows the $\ell_2$ norms of the gradients of the discriminator with respect to the real data points. The closer to 1 the norms are, the better the 1-Lipschitz continuity is preserved. Figure 8 further demonstrates that our consistency (CT) regularization is able to improve GP-WGAN (Gulrajani et al., 2017).

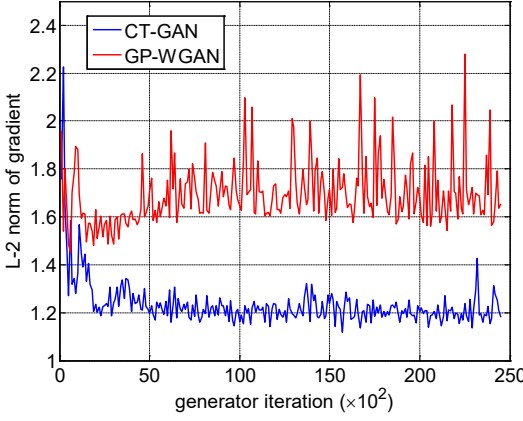

Figure 8: The maximum $\ell_2$ norm of the gradients of the discriminator with respect to the input on CIFAR-10 testing set in each iteration of the training using 1000 CIFAR-10 training images.

**Definition of the Lipschitz continuity.**  Additionally, we check how much the 1-Lipschitz continuity is satisfied according to its vanilla definition (cf. Eq. (3)). Figures 9 and 10 plot the CTs of Eq. (4) and Eq. (5), respectively, over different iterations of the training using 1000 CIFAR-10 images. Figure 10 is the actual CTs used to train the generative model. Figure 9 is drawn as follows.

For every 100 iterations, we randomly pick up 64 real examples and split them into two subsets of the same size. We compute $d(D(x_1) - D(x_2))/d(x_1 - x_2)$ for all the $(x_1, x_2)$ pairs, where $x_1$ is from the first subset and $x_2$ is from the second. The maximum of $d(D(x_1) - D(x_2))/d(x_1 - x_2)$ is plotted in Figure 9. We can see that the CT-GAN curve converges under a certain value much faster than GP-WGAN. The 1-Lipschitz continuity is better maintained by CT-GAN than GP-WGAN over the whole course of the training procedure on the manifold of the real data.

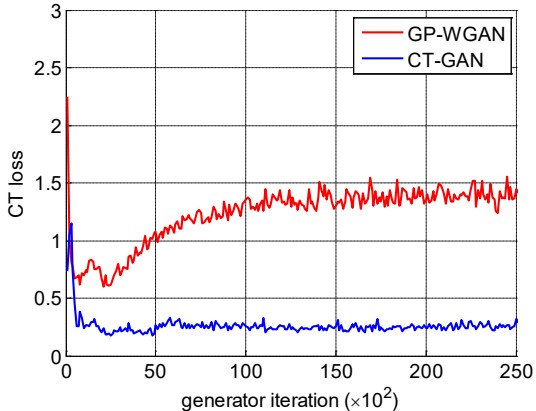

Figure 9: CT (cf. Eq. (4)) over different iterations of the training using 1000 CIFAR-10 images. In each iteration, we randomly pick up two real data points to compute the CT.

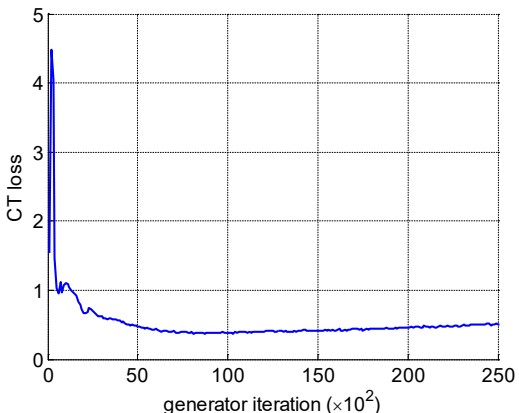

Figure 10: CT (cf. Eq. (5)) over different iterations of the training using 1000 CIFAR-10 images.

## APPENDIX E   GP-WGAN WITH DROPOUT

We run another experiment of GP-WGAN by adding the same dropout layers used in our method to the networks of GP-WGAN, denoted by GP-WGAN+dropout. This may help understand the contribution of our approach in preventing the overfitting problem — is it due to the CT regularization, or merely due to the dropout? This experiment is done by removing the CT term out of our value function for training WGAN using 1,000 CIFAR-10 images and yet keep the dropout layers.

The Inception score of GP-WGAN+dropout is $4.29 \pm 0.12$, and the generated samples are shown in Figure 11(a). We also plot the curve of CT in the training procedure of GP-WGAN+dropout and contrast it to the curve of our CT-GAN in Figure 11(b). It is clear that GP-WGAN+dropout outperforms GP-WGAN ($2.98 \pm 0.11$), and our method ($5.13 \pm 0.12$) outperforms GP-WGAN with a large margin.

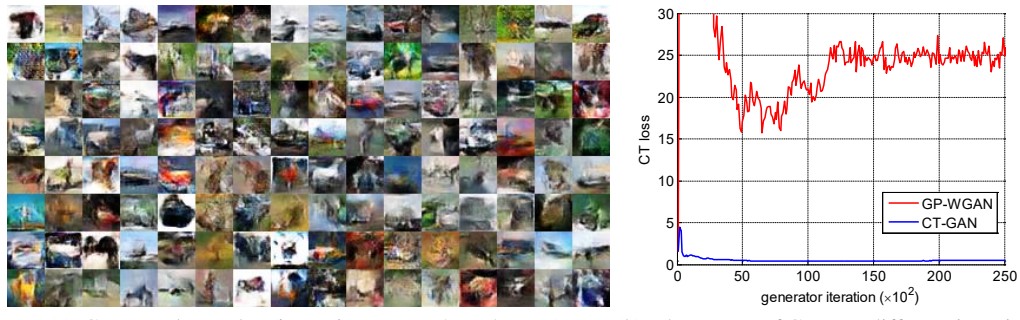

(a) Generated samples (inception score: $4.09 \pm 0.12$)  (b) The curves of CT over different iterations

Figure 11: The results of GP-WGAN+dropout.

Finally, we plot the convergence curves of the discriminators' negative cost function learned by GP-WGAN, GP-WGAN+Dropout, and our CT-GAN in Figure 12. We can see that dropout is able to reduce the overfitting of GP-WGAN, but it is not as effective as our CT-GAN.

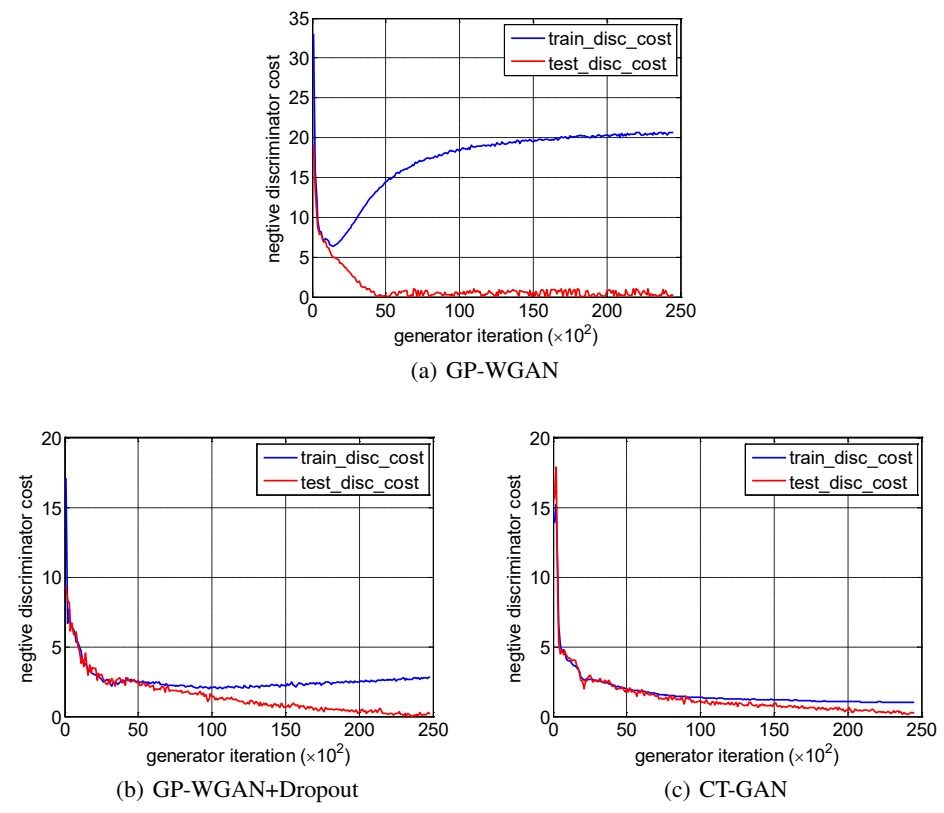

Figure 12: Convergence curves of the discriminator cost: (a) GP-WGAN, (b) GP-WGAN+Dropout, and (c) CT-GAN (ours).

## APPENDIX F  EXPERIMENTS ON LARGE DATASET

In this section, we further present experimental results on the large-scale ImageNet (Deng et al., 2009) and LSUN bedroom (Yu et al., 2015) datasets. The experiment setup (e.g., network architecture, learning rates, etc.) is exactly the same as in the GP-WGAN work (Gulrajani et al., 2017). We

refer readers to (Gulrajani et al., 2017) or our code on GitHub (https://github.com/biuyq/CT-GAN) for the details of the setup.

Our experiment on ImageNet is conducted using images of the size $64 \times 64$. After $200,000$ generator iterations, the inception score of the proposed CT-GAN is **10.27±0.15**, whereas GP-WGAN's is **9.85±0.17**. In addition, the inception score comparison of GP-WGAN and our CT-GAN in each generator iteration is shown in Figure 13. We can observe that the inception score of our proposed CT-GAN becomes higher than GP-WGAN's after early generator iterations. Finally, Figure 14 shows the samples generated by GP-WGAN and CT-GAN, respectively.

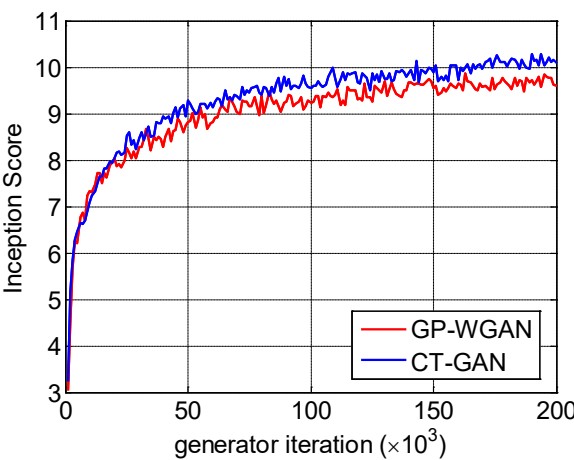

Figure 13: Inception score of GP-WGAN and our CT-GAN with respect to iterations

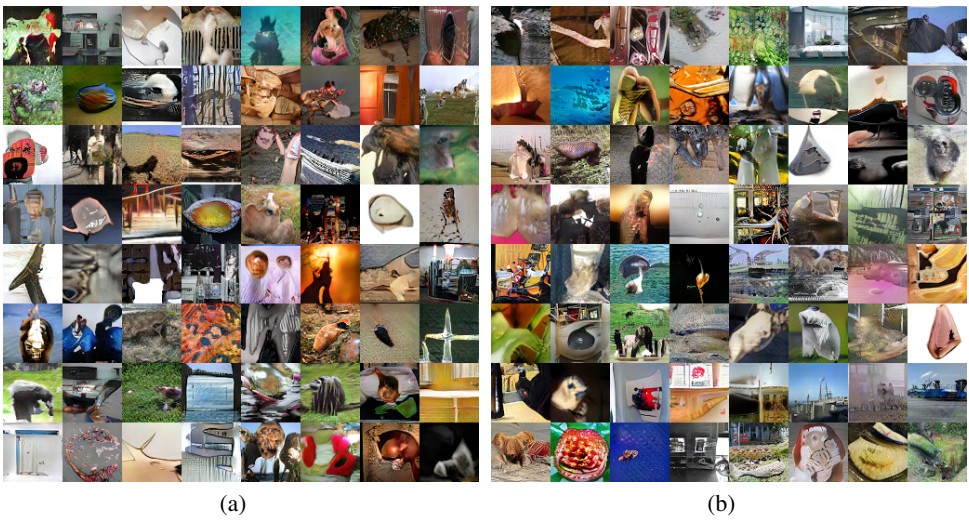

Figure 14: Samples generated by GP-WGAN (a) and by CT-GAN (b), respectively.

For the LSUN bedroom dataset, we show some results in Figure 15. They are the generated samples by our CT-GAN generator after 20k training iterations.

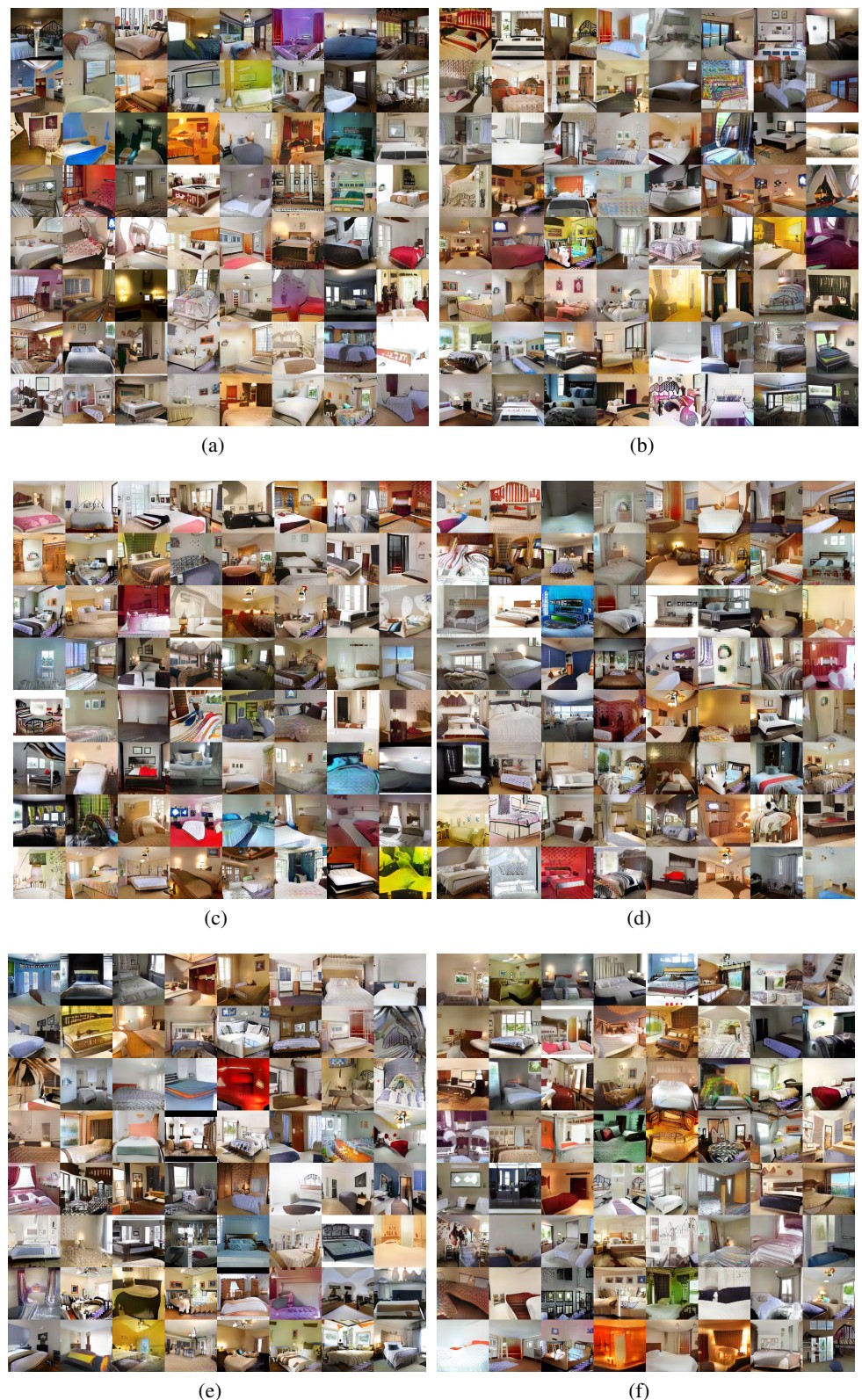

Figure 15: Image samples generated by CT-GAN trained model using LSUN bedroom images.

