# OpenReview forum: "Improving the Improved Training of Wasserstein GANs: A Consistency Term and Its Dual Effect"
_ICLR.cc/2018/Conference — Accept (Poster)_

### Official Review · AnonReviewer3 · 2017-11-26
**Review for "Improving the Improved Training of Wasserstein GANs"**

**Rating:** 6
**Confidence:** 5

**Review:**

Summary:

The paper proposes a new regularizer for wgans, to be combined with the traditional gradient penalty. The theoretical motivation is bleak, and the analysis contains some important mistakes. The results are very good, as noticed by the comments, the fact that the method is also less susceptible to overfitting is also an important result, though this might be purely due to dropout. One of the main problems is that the largest dataset used is CIFAR, which is small. Experiments on something like bedrooms or imagenet would make the paper much stronger.

If the authors fix the theoretical analysis and add evidence in a larger dataset I will raise the score.

Detailed comments:

- The motivation of 1.2 and the sentence "Arguably, it is fairly safe to limit our scope to the manifold that supports the real data distribution P_r and its surrounding regions" are incredibly wrong. First of all, it should be noted that the duality uses 1-Lip in the entire space between Pr and Pg, not in Pr alone. If the manifolds are not extremely close (such as in the beginning of training), then the discriminator can be almost exactly 1 in the real data, and 0 on the fake. Thus the discriminator would be almost exactly constant (0-Lip) near the real manifold, but will fail to be 1-lip in the decision boundary, this is where interpolations fix this issue. See Figure 2 of the wgan paper for example, in this simple example an almost perfect discriminator would have almost 0 penalty.

- In the 'Potential caveats' section, the implication that 1-Lip may not be enforced in non-examined samples is checkable by an easy experiment, which is to look for samples that have gradients of the critic wrt the input with norm > 1. I performed the exp in figure 8 and saw that by taking a slightly higher lambda, one reaches gradients that are as close to 1 as with ct-gan. Since ct-gan uses an extra regularizer, I think the authors need some stronger evidence to support the claim that ct-gan better battles this 'potential caveat'.

- It's important to realize that the CT regularizer with M' = 1 (1-Lip constraint) will only be positive for an almost 1-Lip function if x and x' are sampled when x - x' has a very similar direction than the gradient at x. This is very hard in high dimensional spaces, and when I implemented a CT regularizer indeed the ration of eq (4) was quite less than the norm of the gradient. It would be useful to plot the value of the CT regularizer (the eq 4 version) as the training iterations progresses. Thus the CT regularizer works as an overall Lipschitz penalty, as opposed to penalizing having more than 1 for the Lipschitz constant. This difference is non-trivial and should be discussed.

- Line 11 of the algorithm is missing L^(i) inside the sum.

- One shouldn't use MNIST for anything else than deliberately testing an overfitting problem. Figure 4 is thus relevant, but the semi-supervised results of MNIST or the sample quality experiments give hardly any evidence to support the method.

- The overfitting result is very important, but one should disambiguate this from being due to dropout. Comparing with wgangp + dropout is thus important in this experiment.

- The authors should provide experiments in at least one larger dataset like bedrooms or imagenet (not faces, which is known to be very easy). This would strengthen the paper quite a bit.

---

> ### Author Response · Authors · 2017-12-17
> **Response to ICLR 2018 Conference Paper1144 AnonReviewer3**
>
> We thank the reviewer for the insightful comments and suggestions! The paper has been revised accordingly. Next, we answer the questions in detail.
>
> == Q1: The motivation ==
> We acknowledge that the duality uses 1-Lipschitz continuity in the entire space between Pr and Pg, and it is impossible to visit everywhere of the space in the experiments. We instead focus on the region around the real data manifold to complement the region checked by GP-WGAN --- the gradient penalty term is kept in our overall approach. We have clarified this point by the following in the revised paper.
>
> Arguably, it is fairly safe to limit our scope to the manifold that supports the real data distribution $\mathbb{P}_r$ and its surrounding regions mainly for two reasons. First, we  keep the gradient penalty term  and improve it by the proposed consistency term in our overall approach. While the former enforces the continuity over the points sampled between the real and generated points, the latter complement the former by focusing on the region around the real data manifold instead. Second, the distribution of the generative model $\mathbb{P}_G$ is virtually desired to be as close as possible to $\mathbb{P}_r$.
>
>
> == Q2: That 1-Lip may not be enforced in non-examined samples is checkable ==
> The non-examined samples can refer to all the possible samples in the continuous space which cannot be traversed in a discrete manner. Figure 8 plots the norm of the gradients (of the critic with respect to the input) over the real data points only. In other words, Figure 8 is only part of the consequence, and certainly not the cause, of the discriminators trained by GP-WGAN and our CT-GAN, respectively. It is not surprising that the norms by CT-GAN are closer to 1 than by GP-WGAN because we explicitly enforce the continuity around the real data.
>
> We have run more experiments with larger \lambda values in GP-GAN, and found the gradient norms can indeed reach those of CT-GAN when the \lambda is four times larger than the original one used in the authors’ code. However, the inception score on CIFAR-10 drops a little, and the overfitting remains.
>
> Stronger evidence? In addition to the gradient norm, we have also examined the 1-Lipschitz continuity of the critic using the basic definition. For any two inputs x and x', the difference of the critic's outputs should be no more than M*|x-x'|. This notion is captured by our CT term defined in eq. (4). We plot the CT versus the training iterations as Figure 9 in the revised paper. In particular, for every 100 iterations, we randomly pick up 64 real examples and split them into two subsets of the same size. We compute d(D(x1)-D(x2))/d(x1-x2) for all the (x1,x2) pairs, where x1 is from the first subset and x2 is from the second. The maximum of d(D(x1)-D(x2))/d(x1-x2) is plotted in Figure 9. We can see that the CT-GAN curve converges under a certain value much faster than GP-WGAN.
>
> ==  Q3: Plot the value of the CT regularizer ==
> Please see Figures 9 and 10 in the revised paper for the plots. Note that M’ has absorbed the term d(x’,x’’) in the final consistency term (eq. (5)), so we have to tune its value as opposed to fixing it to 1. Also, because of this fact, we agree with the comment that “Thus the CT regularizer works as an overall Lipschitz penalty, as opposed to penalizing having more than 1 for the Lipschitz constant.” We will clarify this part in the final paper, if it is accepted.
>
> == Q4: Line 11 ==
> It is correct and is another way of denoting the gradient.
>
> == Q5: MNIST ==
> We understand your concern with the use of MNIST and appreciate that you agree the overfitting experiments (Figure 4) are relevant. The other results (e.g., the generated samples and the test error in semi-supervised learning) can give the readers a concrete understanding about our model, but we agree one should not use MNIST to compare different algorithms.
>
>
> == Q6: GP-WGAN + Dropout ==
> Please see Appendix E for the experimental results of GP-WGAN+Dropout on CIFAR-10 using 1000 training images. The corresponding inception score is better than GP-WGAN and yet still significantly lower than ours (2.98+-0.11 vs. 4.29+-0.12 vs. 5.13+-0.12). Figure 12, which is about the convergence curves of the discriminator cost over both training and testing sets, shows that dropout is indeed able to reduce the overfitting, but it is not as effective as ours.
>
>
> == Q7: Experiments in larger datasets ==
> In Appendix F of the revised paper, we present results on the ImageNet and LSUN bedroom datasets following the experiment setup of GP-WGAN. After 200,000 generator iterations on ImageNet, the inception score of CT-GAN is 10.27+-0.15, whereas GP-WGAN's is 9.85+-0.17. Since there is only one class in LSUN bedroom, the inception score is not a proper evaluation metric for the experiments on this dataset. Visually, there is no clear difference between the generated samples of GP-WGAN and CT-GAN up to the 124,000th generator iteration.

---

> > ### Comment · AnonReviewer3 · 2018-01-12
> > **Rebuttal response**
> >
> > Thank you for the rebuttal! I'm glad you took the comments into account and updated the manuscript.
> >
> > A few big picture comments and conclusions, and why I decided to keep my score based on rebuttal, revision and newer comments:
> > - I still think the justification of *why* the regularizer should give an increase in performance is not on the level of a venue like this, and most of the arguments on the theoretical motivation are not strong. I don't find the rebuttal sufficient in this context. If keeping the gradient penalty makes it safe to only check the data manifold, it's still not clear why your method should give an improvement over it (for example, what's a solid theoretical argument for saying enforcing 1-lip in the data manifold more important than elsewhere?).
> > - The authors did a good job at showing improvement in their models on the CIFAR dataset. However, this is not sufficient evidence to prove scalability of the regularizer, especially considering that adding hyperparameters and tuning well in a small dataset like CIFAR can give drastically different results. Furthermore, the results in LSUN and Imagenet are very unconvincing.
> > - I appreciate Appendices D and E. These are very important sanity checks and I'm glad you added them.
> >
> > I think the paper is in a better state now, but taking all of these things into account, the given score is accurate in my opinion.

---

> > > ### Author Response · Authors · 2018-01-22
> > > **Experiments on LSUN finished**
> > >
> > > FYI, we have finally finished the experiments on LSUN-Bedroom. The results are comparable to those reported in (Gulrajani et al. 2017) except that our generated images are more diverse in terms of the color theme.
> > >
> > > 1. https://goo.gl/MvK2x8
> > > 2. https://goo.gl/Cidqgu
> > > 3. https://goo.gl/f6WMeJ
> > > 4. https://goo.gl/N3Jc6M
> > > 5. https://goo.gl/XCpmaK

---

### Official Review · AnonReviewer1 · 2017-11-27
**The paper continues a line of improvement to Wasserstein GANs, and suggests an approach based a double perturbation of each data point, penalizing deviations from Lipshitz-ness. Empirical results demonstrate the effectiveness of the proposal.**

**Rating:** 7
**Confidence:** 4

**Review:**

This paper continues a trend of incremental improvements to Wasserstein GANs (WGAN), where the latter were proposed in order to alleviate the difficulties encountered in training GANs. Originally, Arjovsky et al.  [1] argued that the Wasserstein distance was superior to many others typically used for GANs. An important feature of WGANs is the requirement for the discriminator to be 1-Lipschitz, which [1] achieved simply by clipping the network weights. Recently, Gulrajani et al. [2] proposed a gradient penalty "encouraging" the discriminator to be 1-Lipschitz. However, their approach estimated continuity on points between the generated and the real samples, and thus could fail to guarantee Lipschitz-ness at the early training stages. The paper under review overcomes this drawback by estimating the continuity on perturbations of the real samples. Together with various technical improvements, this leads to state-of-the-art practical performance both in terms of generated images and in semi-supervised learning.

In terms of novelty, the paper provides one core conceptual idea followed by several tweaks aimed at improving the practical performance of GANs. The key conceptual idea is to perturb each data point twice and use a Lipschitz constant to bound the difference in the discriminator’s response on the perturbed points.  The proposed method is used in eq. (6) together with the gradient penalty from [2]. The authors found that directly perturbing the data with Gaussian noise led to inferior results and therefore propose to perturb the hidden layers using dropout. For supervised learning they demonstrate less overfitting for both MNIST and CIFAR 10.  They also extend their framework to the semi-supervised setting of Salismans et al 2016 and report improved image generation.

The authors do an excellent comparative job in presenting their experiments. They compare numerous techniques (e.g., Gaussian noise, dropout) and demonstrates the applicability of the approach for a wide range of tasks. They use several criteria to evaluate their performance (images, inception score, semi-supervised learning, overfitting, weight histogram) and compare against a wide range of competing papers.

Where the paper could perhaps be slightly improved is writing clarity. In particular, the discussion of M and M' is vital to the point of the paper, but could be written in a more transparent manner. The same goes for the semi-supervised experiment details and the CIFAR-10 augmentation process. Finally, the title seems uninformative. Almost all progress is incremental, and the authors modestly give credit to both [1] and [2], but the title is neither memorable nor useful in expressing the novel idea.
[1] Martin Arjovsky, Soumith Chintala, and Leon Bottou. Wasserstein gan.

[2] Ishaan Gulrajani, Faruk Ahmed, Martin Arjovsky, Vincent Dumoulin, and Aaron Courville. Improved training of wasserstein gans.

---

> ### Author Response · Authors · 2017-12-17
> **Response to ICLR 2018 Conference Paper1144 AnonReviewer1**
>
> We thank the reviewer for the very positive and affirmative comments about our work.
>
> We also appreciate the suggestions for improving the writing clarify of the paper. The following has been incorporated in the revised paper.
>
> == M vs. M' ==
> We use the notation $M$ in eq. (3) and a different $M'$ in eq. (4) to reflect the fact that the continuity will be checked only sparsely at some data points in practice. ... ... Note that, however, it becomes impossible to compute the distance $d(\bm{x}',\bm{x}'')$ between the two virtual data points. In this work, we assume it is bounded by a constant and absorb the constant to $M'$. Accordingly, we tune $M'$ in our experiments to take account of this unknown constant; the best results are obtained between $M'=0$ and $M'=0.2$.
>
> == Semi-supervised experiment details and the CIFAR-10 augmentation process ==
>
> MNIST: There are 60,000 images in total. We randomly choose 10 data points for each digit as the labeled set. No data augmentation is used.
>
> CIFAR-10: There are 50,000 image in total. We randomly choose 400 images for each class as the labeled set. We augment the data by horizontally flipping the images and randomly translating the images within [-2,2] pixels. No ZCA whitening is used.
>
> Model Configuration
>
> Table 1: MNIST
> --------------
> Classifier C                        | Generator G
> Input: Labels y, 28*28 Images x     | Input: Noise 100 z
> Gaussian noise 0.3, MLP 1000, ReLU  | MLP 500, Softplus, Batch norm
> Gaussian noise 0.5, MLP 500, ReLU   | MLP 500, Softplus, Batch norm
> Gaussian noise 0.5, MLP 250, ReLU   | MLP 784, Sigmoid, Weight norm
> Gaussian noise 0.5, MLP 250, ReLU   |
> Gaussian noise 0.5, MLP 250, ReLU   |
> Gaussian noise 0.5, MLP 10, Softmax |
>
> Table 2: CIFAR-10
> -----------------
> Input: Labels y, 32*32*3 Colored Image x,               |   Input: Noise 50 z
> ------------------------------------------------------------------------------
> 0.2 Dropout                                             |   MLP 8192, ReLU, BN
> 3*3 conv. 128, Pad =1, Stride =1, lReLU, Weight norm    |   Reshape 512*4*4
> 3*3 conv. 128, Pad =1, Stride =1, lReLU, Weight norm    |   5*5 deconv. 256*8*8,
> 3*3 conv. 128, Pad =1, Stride =2, lReLU, Weight norm    |   ReLU, Batch norm
> ------------------------------------------------------------------------------
> 0.5 Dropout                                             |
> 3*3 conv. 256, Pad =1, Stride =1, lReLU, Weight norm    |
> 3*3 conv. 256, Pad =1, Stride =1, lReLU, Weight norm    |   5*5 deconv. 128*16*16,
> 3*3 conv. 256, Pad =1, Stride =2, lReLU, Weight norm    |   ReLU, Batch norm
> ------------------------------------------------------------------------------
> 0.5 Dropout                                             |
> 3*3 conv. 512, Pad =0, Stride =1, lReLU, Weight norm    |
> 3*3 conv. 256, Pad =0, Stride =1, lReLU, Weight norm    |   5*5 deconv. 3*32*32,
> 3*3 conv. 128, Pad =0, Stride =1, lReLU, Weight norm    |   Tanh, Weight norm
> -----------------------------------------------------------------------------
> Global pool                                             |
> MLP 10, Weight norm, Softmax                            |
>
> == Hyper-parameters ==
> We set \lambda = 1.0 in Eq.(7) in all our experiments. For CIFAR-10, the number of training epochs is set to 1,000 with a constant learning rate of 0.0003. For MNIST, the number of training epochs is set to 300 with a constant learning rate of 0.003.  The other hyper-parameters are exactly the same as in the improved GAN (Salimans et al., 2016).
>
> == New Title ==
> Improving the Improved Training of Wasserstein GANs: A Consistency Term and Its Dual Effect

---

### Official Review · AnonReviewer2 · 2017-11-28

**Rating:** 4
**Confidence:** 4

**Review:**

Updates: thanks for the authors' hard rebuttal work, which addressed some of my problems/concerns. But still, without the analysis of the temporal ensembling trick [Samuli & Timo, 2017] and data augmentation, it is difficult to figure out the real effectiveness of the proposed GAN. I would insist my previous argument and score.

Original review:
-----------------------------------------------------------------------------------------------------------------------------------------------------------------------
This paper presented an improved approach for training WGANs, by applying some Lipschitz constraint close to the real manifold in the pixel level.  The framework can also be integrated to boost the SSL performances. In experiments, the generated data showed very good qualities, measured by inception score. Meanwhile, the SSL-GANs results were impressive on MNIST and CIFAR-10, demonstrating its effectiveness.

However, the paper has the following weakness:

Missing citations: the most related work of this one is the DRAGAN work. However, it did not cite it. I think the author should cite it, make a clear justification for the comparison and emphasize the main contribution of the method. Also, it suggested that the paper should discuss its relation to other important work, [Arjovsky & Bottou 2017], [Wu et al. 2016].

Experiments: as for the experimental part, it is not solid. Firstly, although the SSL results are very good, it is guaranteed the proposed GAN is good [Dai & Almahairi, et al. 2017]. Secondly, the paper missed several details, such as settings, model configuration, hyper-parameters, making it is difficult to justify which part of the model works. Since the paper using the temporal ensembling trick [Samuli & Timo, 2017],  most of the gain might be from there. Data augmentation might also help to improve. Finally, except CIFAR-10, it is better to evaluate it on more datasets.

Given the above reason, I think this paper is not ready to be published in ICLR. The author can submit it to the workshop and prepare for next conference.

---

> ### Author Response · Authors · 2017-12-17
> **Response to ICLR 2018 Conference Paper1144 AnonReviewer2**
>
> We are pleased to see that the reviewer thinks our "generated data showed very good qualities" and "the SSL-GANs results were impressive".
>
> =Q1=
> "Missing citations: the most related work of this one is the DRAGAN"
>
> We would consider WGAN and WGAN-GP as the most related works to ours and DRAGAN ranks after them. As a matter of fact, DRAGAN is an unpublished work and has not been peer-reviewed. As another matter of fact, the gradient penalty in DRAGAN is the same as in WGAN-GP except that it is imposed around the real data while WGAN-GP applies it to the points sampled between the real and the generated ones.
>
> Next, we highlight some key differences between DRAGAN and ours.
>
> We propose to improve Wasserstein GAN, while DRAGAN works with GAN.
>
> DRAGAN aims to reduce the non-optimal saddle points in the minmax two-player training of GANs. In contrast, we propose an approach to enforcing the 1-Lipschitz continuity over the critic of WGANs.
>
> One of our key observations is that it blurs the generated samples if we add noise directly to the data points, as done in DRAGAN. Instead, we perturb the hidden layers of the discriminator.
>
> DRAGAN perturbs a data point once while we do it twice in each iteration. After the perturbations, DRAGAN penalizes the gradients while we enforce the consistency of the outputs.
>
> One of the most distinct features of our approach is that it seamlessly integrates the semi-supervised learning method by Laine & Aila (2016) with GANs.
>
> =Q2=
> "the paper should discuss ... [Arjovsky & Bottou 2017], [Wu et al. 2016]"
>
> We had included both in our paper. Arjovsky & Bottou 2017 analyzes some distribution divergences and their effects in training GANs. Wu et al. 2016 propose to quantitatively evaluate the decoder-based generative models by annealed importance sampling. In our paper, we focus on a different subject, i.e., to design an algorithmic solution to the difficulty of training GANs.
>
> =Q3=
> "the paper missed several details"
>
> Please see either Appendices A and B of the revised paper or the following for our answer to this question.
>
> Given the context of the question, we believe it is about SSL. We follow the experiment setups in the prior works so that our results are directly comparable to theirs. Please see below for more details. If you are interested, you may also check out our code: https://github.com/biuyq/CT-GAN/blob/master/CT-GANs/Theano_classifier/CT_CIFAR-10_TE.py.
>
> MNIST: There are 60,000 images in total. We randomly choose 10 data points for each digit as the labeled set. No data augmentation is used.
>
> CIFAR-10: There are 50,000 image in total. We randomly choose 400 images for each class as the labeled set. We augment the data by horizontally flipping the images and randomly translating the images within [-2,2] pixels. No ZCA whitening is used.
>
> Model Configurations: We had included them in the appendix.
>
> Hyper-parameters: We set lambda = 1.0 in Eq.(7) in all our experiments. For CIFAR-10, the number of training epochs is set to 1,000 with a constant learning rate of 0.0003. For MNIST, the number of training epochs is set to 300 with a constant learning rate of 0.003.  The other hyper-parameters are exactly the same as in the improved GAN (Salimans et al., 2016).
>
> =Q4=
> "... which part of the model works"
>
> Please see either Appendix C of the revised paper or the following for our answer to this question.
>
> We have done some ablation studies about our semi-supervised learning approach on CIFAR-10.
> Method, Error
> w/o CT, 15.0
> w/o GAN (note 1), 12.0
> w batch norm (note 2), --
> w/o D_, 10.7
> OURS, 10.0
>
> Note 1: This almost reduces to TE (Laine & Aila, 2016). All the settings here are the same as TE except that we use the extra regularization ($D\_(.,.)$ in CT) over the second-to-last layer.
> Note 2: We use the weight normalization as in (Salimans et al., 2016), which becomes a core constituent of our approach. The batch normalization would actually invalidate the feature matching in (Salimans et al., 2016).
>
> We can see that both GAN and the temporal ensembling effectively contribute to our final results. The results without our consistency regularization (w/o CT) drop more than those without GAN. We are running the experiments without any data augmentation and will include the corresponding results in the paper.
>
> =Q5=
> "...it is better to evaluate it on more datasets"
>
> We have run some new experiments on the SVHN dataset. Ours is the best among all the GAN based semi-supervised learning methods, and is on par with the state of the arts.
> Method, Error
> PI Laine & Aila 2016, 4.8
> TE Laine & Aila 2016, 4.4
> Tarvainen & Valpola 2017, 4.0
> Miyato et al. 2017, 3.86
> Salimans et al. 2016, 8.1
> Dumoulin et al. 2016, 7.4
> Kumar et al. 2017, 5.9
> Ours, 4.2

---

> ### Author Response · Authors · 2018-01-22
> **About data augmentation**
>
> Following (Laine & Aila, 2016, Miyato et al., 2017, Tarvainen & Valpola, 2017), we do not apply any augmentation to MNIST and yet augment the CIFAR10 images in the following way. We flip the images horizontally and randomly translate the images within [-2,2] pixels horizontally.
>
> Samuli Laine and Timo Aila.  Temporal ensembling for semi-supervised learning. arXiv preprint arXiv:1610.02242, 2016.
> Takeru Miyato, Shin-ichi Maeda, Masanori Koyama, and Shin Ishii. Virtual adversarial training: a regularization method for supervised and semi-supervised learning. arXiv preprint arXiv:1704.03976, 2017.
> Antti Tarvainen and Harri Valpola.  Weight-averaged consistency targets improve semi-supervised deep learning results. arXiv preprint arXiv:1703.01780, 2017.

---

> ### Author Response · Authors · 2018-01-22
> **About the analysis of the temporal ensembling trick [Samuli & Timo, 2017]**
>
> AnonReviewer2: "But still, without the analysis of the temporal ensembling trick [Samuli & Timo, 2017]"
>
> We have actually reported the ablation study about this temporal ensemebling technique in the rebuttal. Please read our answer to Q4 in the rebuttal.
>
> =Q4=
> "... which part of the model works"
>
> Please see either Appendix C of the revised paper or the following for our answer to this question.
>
> We have done some ablation studies about our semi-supervised learning approach on CIFAR-10.
> Method, Error
> w/o CT, 15.0
> w/o GAN (note 1), 12.0
> w batch norm (note 2), --
> w/o D_, 10.7
> OURS, 10.0
>
> Note 1: This almost reduces to TE (Laine & Aila, 2016). All the settings here are the same as TE except that we use the extra regularization ($D\_(.,.)$ in CT) over the second-to-last layer.
> Note 2: We use the weight normalization as in (Salimans et al., 2016), which becomes a core constituent of our approach. The batch normalization would actually invalidate the feature matching in (Salimans et al., 2016).
>
> We can see that both GAN and the temporal ensembling effectively contribute to our final results. The results without our consistency regularization (w/o CT) drop more than those without GAN. We are running the experiments without any data augmentation and will include the corresponding results in the paper.

---

### Public Comment · (anonymous) · 2017-10-29
**Missing Experiment Parameters**

I appreciate your contribution to generative model research. The results seem to be great.
I was trying to reproduce the paper's result, but I think it is difficult to get some details:
(1) Dropout ratio for generative models is not specified. It might be better to have Tables like Tables 4 and 5 for generative modeling tasks.
(2) I could not understand the meaning of "We find that it slightly improves the performance by further controlling the second-to-last layer D_(.) of the discriminator." Are we generating two more perturbed points x''' and x'''' by inserting a dropout layer at second-to-last layer -- as opposed to perturbed points x' and x''?

---

> ### Author Response · Authors · 2017-10-30
> **More experiment details**
>
> Thank you for the interest in our work. Please see below for the answers to your questions.
>
> (1) Dropout ratio:
>
> ## The network used to learn from 1000 labeled CIFAR10 images only
> Discriminator                                                                             Generator
> Input: 3*32*32 Image x                                                            Input: Noise z 128
> 5*5 conv. 128, Pad = same, Stride = 2, lReLU                        MLP 8192, ReLU, Batch norm
> 0.5 Dropout                                                                               Reshape     512*4*4
> 5*5 conv. 256, Pad = same, Stride = 2, lReLU                       5*5 deconv. 256*8*8,
> 0.5 Dropout                                                                              ReLU, Bach norm
> 5*5 conv. 512, Pad = same Stride = 2, lReLU                       5*5 deconv. 128*16*16
> 0.5 Dropout                                                                              ReLU, Batch norm
> Reshape 512*4*4      (D_)                                                        5*5 deconv. 3*32*32
> MLP 1                          (D)                                                         Tanh
>
> ## ResNet:
> Discriminator                                                           |                Generator
> Input: 3*32*32 Image x                                          |                Input: Noise z 128
> [3*3]*2 Residual Block, Resample = DOWN        |                MLP 2048
> 128*16*16                                                               |                Reshape 128*4*4
> [3*3]*2 Residual Block, Resample = DOWN        |                [3*3]*2 Residual Block, Resample = UP
> 128*8*8 0.2 Dropout                                             |                 128*8*8
> [3*3]*2 Residual Block, Resample = None           |                 [3*3]*2 Residual Block, Resample = UP
> 128*8*8 0.5 Dropout                                             |                 128*16*16
> [3*3]*2 Residual Block, Resample = None           |                 [3*3]*2 Residual Block, Resample = UP
> 128*8*8 0.5 Dropout                                             |                 128*32*32
> ReLU, Global mean pool    (D_)                              |                 3*3 conv.  3*32*32
> MLP 1                                  (D)                                |                 Tanh
>
> ## The network for MNIST
> Discriminator                                                                               Generator
> Input: 1*28*28 Image x                                                              Input: Noise z 128
> 5*5 conv. 64, Pad = same, Stride = 2, lReLU                            MLP 4096, ReLU
> 0.5 Dropout                                                                                 Reshape 256*4*4
> 5*5 conv. 128, Pad = same, Stride = 2, lReLU                         5*5 deconv. 128*8*8
> 0.5 Dropout                                                                                 ReLU, Cut 128*7*7
> 5*5 conv. 256, Pad = same, Stride = 2, lReLU                         5*5 deconv. 64*14*14
> 0.5 Dropout                                                                                 ReLU
> Reshape 256*4*4       (D_)                                                          5*5 deconv. 1*28*28
> MLP 1                         (D)                                                             Sigmoid
>
> (2) No. There are only two perturbations, as denoted by x' and x’’, for a data point x in each iteration. They are independently generated by the dropout as shown in my answer to your question (1). In other words, the two terms equation (5) are actually calculated over the same pair of x' and x'' for each draw x ~ P_r.
>
> We will try to release the code in one or two weeks.

---

> > ### Public Comment · (anonymous) · 2017-10-30
> > **Thank you for your reply.**
> >
> > Now I can understand the experimental details. Thanks.

---

### Public Comment · (anonymous) · 2017-10-29
**Smoothing the discriminator near data manifold using local perturbations sounds familiar**

Your contribution looks like a relaxed version of DRAGAN's regularization scheme, which you don't cite anywhere. Is that correct?

Kodali, N., Abernethy, J., Hays, J. and Kira, Z., 2017. How to Train Your DRAGAN. arXiv preprint arXiv:1705.07215.

---

> ### Author Response · Authors · 2017-10-30
> **Thank you for directing us to DRAGAN**
>
> Thank you for directing us to DRAGAN. Sorry for missing it in our paper. We will include it in the updated version.
>
> Going back to your question, the short answer is no because we do not actually aim to smooth the discriminator though the approach may have that effect. The long answer below clarifies it further and additionally highlights some differences between ours and DRAGAN.
>
> Motivations: DRAGAN aims to reduce the non-optimal saddle points in the minmax two-player training of GANs by drawing results from the minimax theorem for zero-sum game. In sharp contrast, we propose an alternative way of enforcing the 1-Lipschitz continuity over the “critic” of WGANs thanks to the recent results by Arjovsky & Bottou (2017).
>
> How to add the perturbations: One of the key observations in our experiments is that it reduces the quality of the generated samples if we add noise directly to the data points, as what is done in DRAGAN. Similar observations are reported by Arjovsky & Bottou (2017) and Wu et al. (2016). After many painstaking trials, we find good results by perturbing the hidden layers of the discriminator instead (as opposed to perturbing the original data). Besides, DRAGAN perturbs a data point once while we do it twice in an iteration.
>
> How to use the perturbation: Similar to the gradient penalty proposed in (Gulrajani et al., 2017), DRGAN introduces a same regularization whereas for different reasons. In contrast, ours is a consistent regularization derived from the basic definition of Lipschitz continuous functions.
>
> Semi-supervised learning: One of the most notable features of our approach is that it seamlessly integrates the semi-supervised learning method by Laine & Aila (2016) with GANs.
>
> Finally, here is the DRAGAN paper we found on ArXiv: https://arxiv.org/abs/1705.07215 just to confirm it with you. Going back to the DRGAN work, it would be interesting to investigate whether it generates blurry images too, for example by comparing the results of different amount of noise including no noise. It may do not because it constraints the gradient as oppose to the discriminator’s output.

---

> > ### Public Comment · (anonymous) · 2017-10-31
> > **Thanks for the clarification**
> >
> > Thanks for the clarification. Comparing with DRAGAN in your experiments would have helped to understand where the benefit is coming from. They show improvements over WGAN-GP as well but use only DCGAN architecture.
> >
> > Good luck!

---

### Public Comment · ~Antti_Tarvainen1 · 2017-10-30
**Citation typo**

Looks like an interesting paper!

I noticed you accidentally cited Salimans et al. on the fourth row of Table 2 when you (probably) meant to cite our work: https://arxiv.org/abs/1703.01780

---

> ### Author Response · Authors · 2017-10-30
> **Thank you and we will correct it in the updated version**
>
> Sorry about that and Thank you for noting it! We will correct it in the updated version.

---

### Public Comment · ~Hongyi_Zhang1 · 2017-11-03
**Any data augmentation?**

In Figure 2, there is an "augmentation" process before feeding the input x into the network D. Could you clarify what "augmentation" means here? In particular, what kind of data preprocessing did you use in the semi-supervised learning experiments?

Thanks!

---

> ### Public Comment · (anonymous) · 2017-11-06
> **Thank you for your concern.**
>
> Dear Hongyi,
>
> Sorry for the late response. We did not receive or had missed the notification from Openreview about your comment. Following (Laine & Aila, 2016, Miyato et al., 2017, Tarvainen & Valpola, 2017), we do not apply any augmentation to MNIST and yet augment the CIFAR10 images in the following way. We flip the images horizontally and randomly translate the images within [-2,2] pixels horizontally.
>
> Samuli Laine and Timo Aila.  Temporal ensembling for semi-supervised learning. arXiv preprint arXiv:1610.02242, 2016.
> Takeru Miyato, Shin-ichi Maeda, Masanori Koyama, and Shin Ishii. Virtual adversarial training: a regularization method for supervised and semi-supervised learning. arXiv preprint arXiv:1704.03976, 2017.
> Antti Tarvainen and Harri Valpola.  Weight-averaged consistency targets improve semi-supervised deep learning results. arXiv preprint arXiv:1703.01780, 2017.

---

> > ### Public Comment · ~Hongyi_Zhang1 · 2017-11-07
> > **Thanks (and additional comments)**
> >
> > Thanks for your clarification, it's very helpful.
> >
> > I think it is good to explicitly compare the data augmentation used by different methods in Table 2, so that the interested readers don't assume they all use the same augmentation, or don't have to look up each paper to figure out what augmentation each method used. For example, AFAIK, the Ladder Networks paper (Table 3, https://arxiv.org/pdf/1507.02672.pdf) reported results without data augmentation.

---

> > > ### Author Response · Authors · 2017-11-08
> > > **Will add a column about the data agumentations**
> > >
> > > Hello Hongyi,
> > >
> > > We will add a column or a new table about the results with and without the data augmentations. Thank you for the pointer!
> > >
> > > Best,

---

### Public Comment · (anonymous) · 2017-11-07
**Source of semi-supervised learning gains**

I'm impressed by your semi-supervised learning results. However, without an ablation study it's hard to tell why your method works so well. Do you have any ideas about what's causing the improvement? It could be
(1) You use both a GAN and consistency regularization (prior work uses one or the other).
(2) Your GAN works better.
(3) Your consistency regularization is better (either because dropout is better than Gaussian noise or because the second-to-last layer consistency term helps).
(4) Improvements to the architecture/hyperparameters (e.g., using weight-norm instead of batch-norm as you mention in the appendix).

---

> ### Author Response · Authors · 2017-11-08
> **Ablated studies**
>
> We have done some ablated studies but they are not as thorough as you suggested. We will complete them and then get back to you soon. Thanks!
>
> Observations thus far: Both the consistent regularization and GAN are necessary to arrive at the report results, and the results without the consistency drop more than those without GAN.

---

> > ### Public Comment · (anonymous) · 2017-11-22
> > **Ablated study results**
> >
> > The table below shows the ablated study results:
> >
> > Method                                                                          |                  Test Error
> > OURS w/o CT                                                               |                 14.98+-0.43
> > OURS w/o GAN *                                                         |                  11.98+-0.32
> > OURS w batch norm **                                                |                           --
> > OURS w/o D_(.,.) over the second-to-last layer        |                  10.70+-0.24
> > OURS                                                                             |                  9.98+-0.21
> >
> > * This almost reduces to TE (Laine & Aila, 2016). All the settings are exactly the same as in TE  (Laine & Aila, 2016) except that we use the extra regularization (D_(.,.) in CT) over the second-to-last layer.
> > ** We use the weight normalization as in (Salimans et al., 2016), which becomes a core constituent of our approach. The batch normalization would actually invalidate the feature matching in  (Salimans et al., 2016).
> >
> > Samuli Laine and Timo Aila. Temporal ensembling for semi-supervised learning. arXiv preprintarXiv:1610.02242, 2016.
> > Tim Salimans, Ian Goodfellow, Wojciech Zaremba, Vicki Cheung, Alec Radford, and Xi Chen.Improved techniques for training gans. In Advances in Neural Information Processing Systems,pp. 2234–2242, 2016.

---

### Public Comment · (anonymous) · 2017-11-16
**Code**

Here is the code for this paper: https://github.com/biuyq/CT-GAN

---

### Public Comment · (anonymous) · 2017-11-28
**Does the intuition agree with what you are doing in the end?**

Section 1.2 and Figure 1 outline the general idea behind the approach.

I wonder however, how much of the intuitive explanation is still valid in the actual CT loss.

To dissect this a little:
d(x1, x2) being a metric should always be positive. that means all instances of max(0, d(x1,x2)) reduce to d(x1,x2)

Looking at Eq 4 and 5, given M=0, d(x1,x2) is assumed to be constant, it reduces to
CT_(x1,x2) = E_{x1,x2} d(D(x1),D(x2))
with the constant d(x1,x2) absorbed into d.

However, since the input is not changed but rather the network, a better notation would probably be

CT = E_(x, \theta_1, \theta_2) d(D(x, \theta_1), D(x, \theta_2))

where \theta_1, \theta_2 are the noise vectors used for dropout.

Looking at that formulation, does this still mean it penalizes the gradient in the original input space or would it be more appropriate to say it encourages the resilience to dropout (or is that actually the same thing)?

---

> ### Author Response · Authors · 2017-12-01
> **The dual roles of that formulation**
>
> That’s a great question! We were actually wondering about a similar one. We can certainly interpret the formulation as that it encourages the network to be resilient to the dropout noise --- one of the notions that motivates the temporal ensembling semi-supervised learning method. In addition to that, however, it also enforces the Lipschitz continuity over the discriminator because of equation (3). Thanks to the dual roles of this formulation, we are able to use it to both improve the training of WGANs and connect GAN with the temporal ensembling method.
>
> We have re-run the experiments using margin $M’=0.2$. To show that the margin, albeit small, plays an active role, we have got some statistics of the $d(D,D)+0.1 d(D_,D_)$ term over the last 10 epochs. We can see that the median values of that term are smaller and the max values are larger than the margin.
>
> Min          0.0162    0.0153    0.0149    0.0171    0.0170    0.0159    0.0140    0.0146    0.0159    0.0144
> Median    0.1130    0.1133     0.1124    0.1138    0.1114      0.1123    0.1124     0.1122    0.1125     0.1111
> Max          7.1718    6.1229    7.1985    7.3505    4.9636    5.2252   5.2559   5.3058    5.9905    4.8519

---

### Public Comment · (anonymous) · 2017-12-16
**Review for paper 1144**

This paper points out a potential caveat for the improved training of WGAN approach. The gradient penalty term takes effects only upon points sampled on the lines connecting pairs of data points sampled from the real distribution and the model distribution. At the beginning of the training the Lipschitz continuity over the manifold supporting the real distribution is not enforced because at the beginning stage the synthetic data points G(z), and hence the sampled points $\hat{x}$, could be far away from the manifold. The author introduces a natural solution to overcome that problem, that is to additionally impose the Lipschitz continuity condition over the manifold supporting the real data distribution. This paper showed that WGAN with consistency term can generate sharper and more realistic samples than most state-of-art GANs. Moreover, they proposed a framework for semi-supervised training which is able to train a decent GAN model.

Generally speaking, the author did a pretty good job on improving the training of WGANs, and the results are very impressive. We re-conducted some of the experiments using the code released by author, and here are several comments based on our findings:

Mathematical rigorousness: Since throughout the experiments M' has been held at 0, it drops out from the consistency term (4); furthermore, since the denominator d(x_1,x_2) is a constant number, and the numerator is by the definition of metrics a value greater than or equal to 0, the consistency term effectively reduces to

     CT|_{x_1,x_2}= E_{x_1,x_2 ~ P_r}[d(D(x_1),D(x_2))].

We find it hard to infer how the Lipschitz continuity is enforced from merely adding a metric d(D(x_1),D(x_2)) as an additional constraint, and we suspect that the actual training has deviated from the initial motivation which was to enforce the Lipschitz continuity over the manifold supporting the real data distribution.

More experiments on larger and higher dimensional dataset: The paper has shown a noticeable improvement on the generated CIFAR-10 image quality. Nevertheless, higher dimensional datasets should be considered in the experiments. This quality improvement on higher dimensional images can be more noticeable. Furthermore, in the paper, the authors only used 1000 samples to train the generator; we think that using larger datasets can also verify their claims more persuasively. We also tried to use the original code from [1] for MNIST data. Our results show that using the whole MNIST dataset leads to less vague digits (low contrast between foreground and background) than only using 1000 images.

Overfitting analysis: The paper stated that CT-WGAN is less prone to overfitting. We have verified this claim in our experiment. However, the reason for that is less clear in the paper. We think the dropout perturbation plays an important role in avoiding overfitting. We modified GP-WGAN architecture by adding the dropout layers in the same way as in CT-WGAN. Our results show that GP-WGAN becomes less prone to overfitting after adding dropout. Therefore, we suspect that the non-overfitting property of CT-WGAN is a direct consequence of adding dropout regularization, instead of adding consistency term.

[1]  I.Gulrajani, F.Ahmed, M.Arjovsky, V.Dumoulin, and A.Courville,  "Improved  training  of  wasserstein  gans,"arXiv  preprintarXiv:1704.00028, 2017.

---

> ### Author Response · Authors · 2017-12-18
> **Thank you for checking out our code!**
>
> Thank you for checking out our code! If you are interested, please also test M’=0.2 for CIFAR-10 and you should be able to see a slightly higher inception score than M’=0. We fix M’=0 for all the experiments in the paper for consistency, but as we wrote in the paper, the best results are obtained between M’=0 and M’=0.2.
>
> We noted that the assumption of d(x_1,x_2) being a constant can be relaxed. Our derivations still hold as long as d(x’,x’’) is bounded by a constant, and we can absorb the constant to M’.
>
> We have actually reported two sets of experiments for CIFAR-10 in the paper. The first set is done using 1000 images and the second uses the whole CIFAR-10 dataset to train a ResNet. These setups are the same as in [1]. Additionally, we are running experiments on ImageNet and LSUN; we will update the response once the experiments are done.
>
> About the overfitting, please see Appendix E of the revised paper for the experimental results of GP-WGAN+Dropout on CIFAR-10 using 1000 training images. The corresponding inception score is better than GP-WGAN and yet still significantly lower than ours (2.98+-0.11 vs. 4.29+-0.12 vs. 5.13+-0.12). Figure 12, which is about the convergence curves of the discriminator cost over both training and testing sets, shows that dropout is indeed able to reduce the overfitting, but it is not as effective as ours.
>
> In Appendix F of the revised paper, we further present  experimental results on the large-scale ImageNet and LSUN bedroom datasets. The experiment setup (e.g., network architecture, learning rates, etc.) is exactly the same as in the GP-WGAN work. After 200,000 generator iterations on ImageNet, the inception score of the proposed CT-GAN is 10.27+-0.15, whereas GP-WGAN's is 9.85+-0.17. Since there is only one class in LSUN bedroom, the inception score is not a proper evaluation metric for the experiments on this dataset. Visually, there is no clear difference between the generated samples of GP-WGAN and CT-GAN.

---

### Decision · Program_Chairs · 2018-01-29
**ICLR 2018 Conference Acceptance Decision**

**Decision:**

Accept (Poster)

**Comment:**

The paper proposes various improvements to Wasserstein distance based GAN training. Reviewers agree that the method produces good quality samples and are impressed by the state of the art results in several semi-supervised learning benchmarks. The paper is well written and the authors have further improved the empirical analysis in the paper based on reviewer comments.